# Fusion of Multiscale Features via Centralized Sparse-attention Network for EEG Motor Imagery Classification

## Abstract

Motor imagery (MI) is an important research direction in brain-computer interfaces (BCIs) and has shown broad application value in motor rehabilitation. In recent years, a number of approaches have leveraged multiscale temporal convolution modules to capture the temporal dynamics of MI data, followed by a unified spatial module to perform spatial feature modeling. However, this design implicitly assumes that all temporal scales share the same spatial structure, overlooking the inherent spatiotemporal heterogeneity of EEG signals. To address this limitation, we design a multi-branch parallel architecture, where each temporal scale is equipped with its own spatial feature extraction module. This design mitigates the risk of spatial information confusion or loss arising from shared weights, while enhancing the flexibility and discriminative capacity of feature representations. Furthermore, to tackle the challenge of multi-branch feature fusion, we introduce the Fusion of Multiscale Features via Centralized Sparse-attention Network (EEG-CSANet). Specifically, EEG-CSANet adopts a main–auxiliary collaborative fusion architecture: the main branch leverages multiscale multi-head self-attention to model core spatiotemporal patterns, while the auxiliary branch employs multiscale sparse cross-attention to achieve efficient local interactions with the main branch. Experimental results demonstrate that EEG-CSANet achieves state-of-the-art (SOTA) performance across three public MI datasets. In particular, it significantly outperforms all compared SOTA methods on the BCI Competition IV 2a and 2b datasets, and also achieves the best results in subject-independent experiments on the 2a dataset. The related code is publicly available at: https://anonymous.4open.science/r/test-tj654478-EB7B

## 1 Introduction

Brain-computer interfaces (BCIs) enable direct communication between the brain and external devices (Wolpaw, 2007). Currently, they have demonstrated broad application prospects in various fields such as human-computer interaction, motor rehabilitation, emotion recognition, disease diagnosis, and treatment (Park et al., 2022); (Zhang et al., 2023); (Edelman et al., 2024). Electroencephalography (EEG) captures voltage fluctuations generated by neural activity in the brain. Due to its advantages of non-invasiveness, high temporal resolution, and good portability, EEG has become the most commonly used signal source in current BCI systems (Liu et al., 2025b).

Motor imagery (MI), as one of the typical paradigms in BCIs, have shown significant value in the field of motor rehabilitation (Huang et al., 2025); (Pfurtscheller & Neuper, 2001); (Leuthardt et al., 2004). Motor imagery-based brain-computer interfaces (MI-BCI) decode users' EEG signals to identify their motor intentions, enabling direct control of external devices without relying on the peripheral nervous system or muscular activity (Meng et al., 2025); (Ang et al., 2015); (Ang et al., 2014). Currently, MI-BCI have been widely applied in driving functional electrical stimulation, intelligent prosthetics, brain-controlled wheelchairs, virtual reality rehabilitation systems, and other assistive rehabilitation devices (Blanco-Diaz et al., 2024); (Alawieh et al., 2025); (Lu et al., 2025), significantly improving therapeutic outcomes and quality of life for patients with neurological disorders such as stroke by promoting neural functional reorganization.

With the rapid development of deep learning technologies, various deep learning models have been widely applied to decode MI tasks. Initially, the introduction of Deep ConvNet, Shallow ConvNet (Schirrmeister et al., 2017), and EEGNet (Lawhern et al., 2018) significantly improved the accuracy and robustness of MI classification, establishing them as mainstream methods for EEG feature extraction. Subsequently, long short-term memory networks (LSTM) (Zhang et al., 2019); (Wang et al., 2018), CNN-LSTM hybrid models (Wang et al., 2023); (Li et al., 2022), and multi-branch CNN architectures (Yang et al., 2021); (Zhao et al., 2019) were successively proposed, further advancing the development of MI decoding techniques. In recent years, with the success of the Transformer architecture in sequence modeling, the self-attention mechanism has been introduced into EEG analysis. Models such as Conformer (Song et al., 2022) and ATCNet (Altaheri et al., 2022) significantly enhanced feature representation capabilities by effectively integrating attention mechanisms with CNN architectures, thereby advancing MI classification accuracy to new levels. Since then, models based on the CNN-Transformer framework have gradually become dominant.

In recent years, numerous outstanding studies have emerged. Models such as MSTFNet (Jin et al., 2024), EEGTransNet (Ma et al., 2024), MCMTNet (Yang et al., 2025), and TMSA-Net (Zhao & Zhu, 2025) have adopted multiscale temporal feature extraction strategies to more effectively uncover deep spatiotemporal patterns in EEG signals. Nevertheless, we posit that current approaches face inherent limitations in the manner in which they fuse spatial and temporal features. Contemporary mainstream architectures typically begin by extracting multiscale temporal features using multiple temporal convolutional kernels of varying sizes, which are then combined via **concatenation** or **addition** before being uniformly fed into subsequent spatial convolutional layers to capture spatial patterns. This design implicitly assumes that all temporal scales share the same spatial structure, thereby overlooking the fact that brain region activation patterns corresponding to different temporal scales may differ significantly. (Li et al., 2025); (Tao et al., 2023) indicate that temporal convolutional kernels of different sizes capture features from different frequency bands, and prior research has demonstrated that brain activation characteristics vary across frequency bands during MI tasks (Liu et al., 2025a); (Chen et al., 2023). Thus, it can be seen that **spatial correlations at different temporal scales exhibit heterogeneous characteristics** in MI task decoding (Wang et al., 2024).

Based on this, we argue that multiscale temporal features should not be simply merged and then processed by a shared spatial filtering module. Alternatively, a **multi-branch parallel architecture** should be adopted, equipping each temporal scale with an independent spatial feature extraction module to enable fine-grained modeling. By doing so, the model can more accurately capture the unique channel coordination patterns at each temporal scale, avoiding the confusion or loss of critical spatial information caused by sharing spatial weights, thereby enhancing the flexibility and discriminative power of feature representation.

However, the effective fusion of multi-branch features remains a key challenge in feature integration. In recent years, cross-attention mechanisms have demonstrated strong fusion potential due to their advantages in modeling inter-branch feature dependencies. Inspired by the cross-attention methods in the works of (Chen et al., 2023); (Liu et al., 2025a); (Ma et al., 2025), **we innovatively propose a fusion architecture based on collaboration between a main branch and auxiliary branches**: the main branch employs a multiscale multi-head self-attention mechanism to extract and enhance core spatio-temporal features, while multiple auxiliary branches separately capture local details at specific scales. During the fusion stage, the auxiliary branches interact with the semantically relevant key local regions in the main branch through multiscale sparse multi-head cross-attention, avoiding the computational redundancy and noise interference caused by global associations. This strategy enables efficient and precise feature aggregation, significantly enhancing the model's representational capacity and robustness.

In summary, the contributions of this paper are as follows:

1. In response to the characteristics of EEG signals, we propose a **multi-branch feature extraction framework** to alleviate the loss of channel discriminative information caused by coarse-grained merging in traditional multiscale temporal feature fusion.

2. We innovatively design a feature fusion architecture with collaboration between a main branch and auxiliary branches: the main branch employs a **multiscale multi-head self-attention mechanism** to enhance modeling of core spatiotemporal patterns, while the auxiliary branches achieve

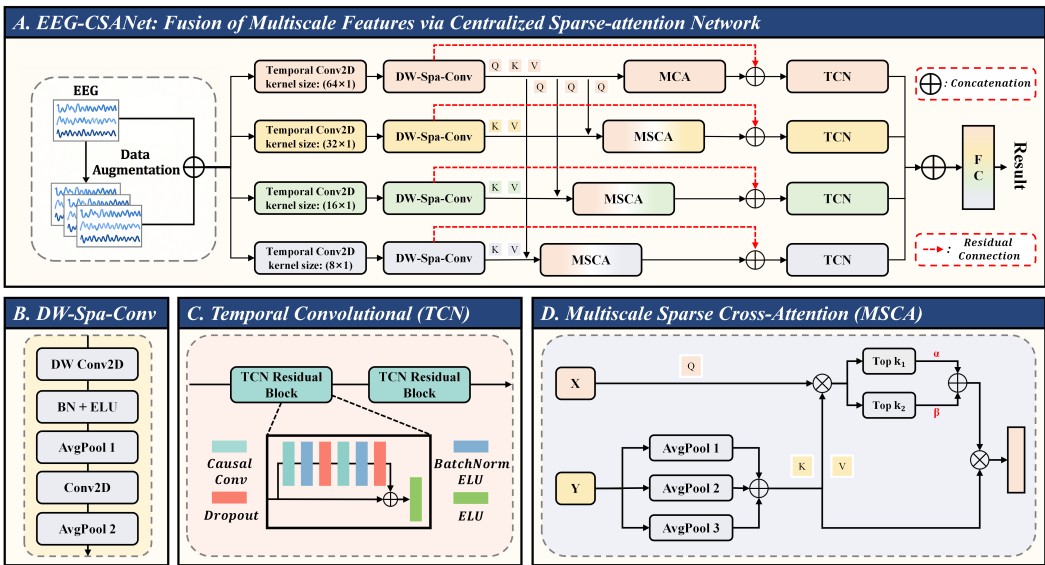

Figure 1: The overall architecture of EEG-CSANet.

efficient and precise feature interaction with key local regions of the main branch through a **multi-scale sparse multi-head cross-attention mechanism**.

3. Experimental results on three public motor imagery datasets demonstrate that the proposed method outperforms existing approaches in classification accuracy, **significantly surpassing all comparison models on the BCI Competition IV 2a and 2b**, thereby validating its effectiveness and robustness.

## 2 METHOD

The proposed EEG-CSANet architecture is illustrated in Figure 1. Initially, a same category signal segmentation and reconstruction (S&R) (Lotte, 2015) method is employed for data augmentation of EEG signals. The original EEG data are then combined with the augmented data to expand the training sample size. Subsequently, the signals are processed by a multiscale temporal feature extraction module. To avoid losing inter-channel interaction information at different scales, a multi-branch structure is adopted, which separately extracts deep channel features for each time scale. To effectively integrate EEG signals extracted from multiple branches, we have innovatively designed a fusion mechanism: the main branch employs a multiscale multi-head self-attention mechanism, while auxiliary branches introduce a multiscale sparse multi-head cross-attention mechanism, achieving efficient information exchange between branches. The integrated features are then fed into a temporal convolutional network to further extract higher-level temporal dynamic features. To prevent the degradation of temporal information within deep networks, residual structures are incorporated into the convolutional modules. Finally, a fully connected layer is used to complete the classification task. The following sections will provide a detailed description of each component of the EEG-CSANet.

**Data augmentation.** We use the method of S&R to perform data augmentation on EEG signals, which is a commonly used data augmentation method in MI (Song et al., 2022); (Ma et al., 2024); (Yang et al., 2025). Specifically, we let $X_i \in \mathbb{R}^{B \times C \times T}$ denote the original EEG signal of the $i$-th class, where $i \in \{1, \ldots, L\}$, $L$ is the total number of classes, and $B$, $C$, and $T$ denote the batch size, number of channels, and number of time samples, respectively. The signal $X_i$ is uniformly divided into $S$ segments along the time dimension. New samples $X_i'$ are then generated by randomly selecting and reorganizing these segments while preserving their original temporal order. This augmentation is applied separately within each training batch, generating an amount of synthetic data equal in size to the original input $X_i$. Therefore, the model input for each batch after augmentation

is given by:

$$X_{\text{Input}} = \text{Concat}\left(X, X'\right), X = \sum_{k=1}^{L} X_i, \tag{1}$$

where $X$ represents the original input signals in the batch, $X_{\text{Input}}$ denotes the complete input fed into the subsequent model, and $\text{Concat}(\cdot)$ denotes concatenation along the first dimension of the data.

**Multi-branch convolution**. Previous studies (Ma et al., 2024); (Yang et al., 2025); (Jin et al., 2024); (Zhao & Zhu, 2025) have tended to employ multiscale temporal convolution fusion methods to extract features from EEG signals, as illustrated in Figure 4. This approach concatenates multiscale temporal features to jointly extract channel-wise features. However, we contend that directly concatenating multiscale temporal features for channel-wise feature extraction may not effectively capture these distinct spatial patterns. Therefore, we propose a multi-branch architecture to separately process and fuse features from different temporal scales, as shown in Figure 1.$A$. In this architecture, each branch is responsible for handling features at a specific temporal scale or frequency band, thereby enabling more accurate capture and representation of the distinct spatial characteristics present across different frequency bands.

The combined structure of temporal convolution and depthwise separable spatial convolution (DW-Spa-Conv) resembles that of EEGNet (Lawhern et al., 2018). Specifically, the input signal $X_{\text{Input}}$ first passes through a temporal convolutional module with kernel size $(K_i, 1)$, where $i \in \{1, 2, 3, 4\}$ corresponds to the four branches, to extract temporal features for each channel. This is followed by a depthwise separable convolution with kernel size $(C, 1)$, where the depth multiplier $D$ controls the number of output feature maps per input channel, enabling spatial feature extraction. Next, the features are downsampled via the first average pooling layer with kernel size $(P_1, 1)$. The pooled features are then fed into a subsequent convolutional module with kernel size $(K_5, 1)$; we regard this module as facilitating further channel-wise interaction, thus enhancing the extraction of high-dimensional spatio-temporal features. Finally, a second average pooling layer with kernel size $(P_2, 1)$ performs additional downsampling. As a result, the output of each of the four convolutional branches is $Z_i \in \mathbb{R}^{B \times U_i \times T_0}$:

$$U_i = F_i \times D, \tag{2}$$

$$T_0 = \frac{T}{P_1 \times P_2} \tag{3}$$

where $F_i$ denotes the number of convolutional kernels in the $i$-th temporal branch, and $T_0$ is the number of time steps after downsampling.

**Feature fusion architecture.** To effectively fuse the features $Z_i$ obtained from the four branches, we propose a feature fusion architecture with collaboration between a main-branch multiscale multi-head self-attention mechanism and multiscale sparse multi-head cross-attention mechanisms for the auxiliary branches.

Specifically, we designate $Z_1$ as the main branch and $Z_2$, $Z_3$, $Z_4$ as auxiliary branches. For the main branch $Z_1$, we adopt a relatively large convolutional kernel $K_1$ for temporal feature extraction, aiming to capture extensive global spatio-temporal dependencies. In contrast, the auxiliary branches $Z_2$, $Z_3$, and $Z_4$ use smaller kernels $K_2$, $K_3$, $K_4$ to preserve finer local spatio-temporal patterns. Due to the constrained receptive fields of small kernels, they are often inadequate for modeling long-range dependencies and can overlook crucial global structural information. To address this issue, we introduce a multiscale sparse multi-head cross-attention mechanism (Chen et al., 2023) that enhances global perception while capturing local fine-grained details, enabling collaborative fusion between the main and auxiliary branches.

As illustrated in Figure 1.D, $X$ and $Y$ are the inputs to the Multiscale Sparse Cross-Attention (MSCA) module. To capture multiscale information from the data, we first apply three average pooling operations with different kernel sizes, and then sum the resulting features to obtain $Y'$:

$$Y' = \sum_{i=1}^{3} Q_i(Y), \quad i \in \{1, 2, 3\}, \tag{4}$$

where $Q_i$ denotes the $i$-th average pooling operation, and $Y'$ denotes the output of the pooling layer. Note that $Y$, $Y_i$, and $Y'$ all have the same spatial dimensions.

We then project the input $X$ into the Query matrix and the transformed input $Y'$ into the Key and Value matrices as follows:

$$Q = XW_q, \quad K = Y'W_k, \quad V = Y'W_v, \tag{5}$$

where $W_q$, $W_k$, and $W_v$ are learnable weight matrices.

The attention score matrix $A$ is subsequently computed using scaled dot-product attention (Vaswani et al., 2017):

$$A = \frac{QK^\top}{\sqrt{d_k}}, \tag{6}$$

where $d_k$ denotes the dimensionality of the key vectors.

Before applying the softmax function, we introduce a Top-$k$ sparsification operation (Chen et al., 2023), which discards the smallest $1 - k$ proportion of values in each row of $A$. This effectively mitigates the influence of noise from auxiliary branches on feature extraction, as illustrated in Figure 1.$D$. For the softmax operation, these discarded values are replaced with $-\infty$, causing them to be transformed into zero after softmax, similar to a selective masking mechanism. The operation is expressed as:

$$A' = \text{softmax}(\text{Top-}k(A)). \tag{7}$$

We apply the Top-$k$ operation with two different ratios, controlled by parameters $k_1$ and $k_2$, and introduce two learnable scalars $\alpha$ and $\beta$ to adaptively balance their contributions, it is a strategy inspired by (Ma et al., 2025). Thus, the final attention output is computed as:

$$\text{Attention} = \alpha \cdot \text{softmax}(\text{Top-}k_1(A)) \cdot V + \beta \cdot \text{softmax}(\text{Top-}k_2(A)) \cdot V. \tag{8}$$

The output of the multi-head attention module is then concatenated across heads:

$$\text{MHA} = \text{Concat}\left(\text{Attention}_0, \text{Attention}_1, \dots, \text{Attention}_{h-1}\right), \tag{9}$$

where $h$ denotes the number of attention heads.

This design enables the effective fusion of information from the main branch $X$ and auxiliary branches $Y$, thereby achieving cross-branch feature enhancement. Notably, sparsification is omitted in the main branch, which instead employs a multiscale multi-head self-attention mechanism to retain global temporal patterns. The combination of multiscale pooling and sparsification, however, may result in a significant loss of temporal information. To address this, we adopt a residual connection (He et al., 2016) to retain the original features:

$$M_i = Z_i + \text{MHA}_i, \tag{10}$$

where $M_i$ is the output of the $i$-th branch, $Z_i$ is the output from the DW-Spa-Conv block, and $\text{MHA}_i$ is the corresponding MHA output.

**Temporal Convolutional.** The TCN structure we use follows the design of (Altaheri et al., 2022), with its core architecture shown in Figure 1.C. This module adopts a double residual structure, aiming to deeply extract high-level temporal features from brain electrical signals. The dilation factors for the two residual blocks are $d_1$ and $d_2$, respectively. Batch normalization and ELU activation functions are then applied to enhance training stability. Finally, after concatenating the data, it is fed into a fully connected layer for classification, yielding the final classification result:

$$\hat{y} = \text{Linear}\left(\text{Concat}(P_1, P_2, P_3, P_4)\right), \tag{11}$$

where Linear represents the final linear classification layer, and $P_i$ represents the output features of the $i$-th layer of the TCN.

## 3 EXPERIMENTS

### 3.1 DATASETS

We consider three publicly available MI-EEG datasets to comprehensively evaluate EEG-CSANet: BCI Competition IV 2a Dataset (BCIC-IV-2a) (Brunner et al., 2008), BCI Competition IV 2b Dataset (BCIC-IV-2b) (Leeb et al., 2008), and the High Gamma Dataset (HGD) (Schirrmeister et al., 2017). The detailed information is listed in Table 1 and Appendix A.1

Table 1: Details of the datasets.

| Dataset | Participants | Sampling rate (Hz) | Channels | Classes |
|---|---|---|---|---|
| BCIC-IV-2a | 9 | 250 | 22 | 4 |
| BCIC-IV-2b | 9 | 250 | 3 | 2 |
| HGD | 14 | 500 | 128 | 4 |

## 3.2 EXPERIMENTAL SETUP

Our experiments were implemented in Python 3.10 using the PyTorch framework, and conducted on an NVIDIA RTX 2080Ti GPU. The number of training epochs was set to 2000 (Ma et al., 2024); (Zhao & Zhu, 2025), with a batch size of 16. During training, the cross-entropy loss function was employed. In terms of network optimization, we used the Adam optimizer (He et al., 2015) with a learning rate of 0.0009. In our subject-independent experiment, we adopted the leave-one-subject-out strategy, using the complete data from eight subjects for training while reserving all data from the ninth subject exclusively for testing.

To ensure a comprehensive evaluation of our model, we adopted two widely recognized classification metrics: accuracy (ACC) and the kappa coefficient. To ensure the reproducibility of model performance, we conducted all experiments with a fixed random seed.

## 3.3 EXPERIMENTAL RESULT

We report two groups of experiments: (1) subject-dependent performance comparisons on BCIC-IV-2a, BCIC-IV-2b, and HGD, and (2) subject-independent performance comparisons on BCIC-IV-2a. To evaluate the robustness of our proposed model, we compared it against the baseline model EEGNet (Lawhern et al., 2018) as well as several state-of-the-art (SOTA) methods published between 2022 and 2025, including EEG-Conformer (Song et al., 2022), ATCNet (Altaheri et al., 2022), ADFCNN (Tao et al., 2023), EEG-TransNet (Ma et al., 2024), MSTFNet (Jin et al., 2024), EISATC-Fusion (Liang et al., 2024), MCMTNet (Yang et al., 2025), and TMSA-Net (Zhao & Zhu, 2025). These SOTA methods are further described in the Appendix A.2.

Table 2: Performance comparison on BCIC-IV-2a.

| Methods (Year) | A1 | A2 | A3 | A4 | A5 | A6 | A7 | A8 | A9 | Acc | Std | Kappa |
|---|---|---|---|---|---|---|---|---|---|---|---|---|
| EEGNet (2018)[***] | 82.91 | 66.49 | 87.29 | 59.39 | 64.88 | 60.59 | 72.81 | 81.06 | 85.57 | 73.44 | 11.02 | 0.6423 |
| EEG-Conformer (2022)[**] | 88.19 | 61.46 | 93.40 | 78.13 | 52.08 | 65.28 | 92.36 | 88.19 | 88.89 | 78.66 | 15.30 | 0.7155 |
| ATCNet (2022)[***] | 85.07 | 68.75 | 96.53 | 83.68 | 77.78 | 72.22 | 86.81 | 86.46 | 89.31 | 82.96 | 8.66 | 0.7840 |
| ADFCNN (2024)[**] | 89.42 | 71.12 | 95.61 | 82.43 | 73.42 | 71.88 | 90.97 | 87.50 | 86.81 | 83.24 | 9.05 | 0.7733 |
| EEG-TransNet (2024)[**] | 88.89 | 64.93 | 96.18 | 85.42 | 82.64 | 73.61 | **95.14** | 90.28 | 88.19 | 85.03 | 10.12 | 0.8004 |
| EISATC-Fusion (2024)[**] | 85.07 | 73.26 | 95.49 | 87.15 | 81.94 | 73.96 | 93.06 | 85.76 | 85.42 | 84.57 | **7.48** | 0.7942 |
| MSTFNet (2024)[**] | 90.63 | 69.89 | 97.22 | 79.56 | 79.86 | 68.40 | 90.97 | 86.11 | 89.93 | 83.62 | 9.91 | 0.7900 |
| MCMTNet (2025)[*] | 89.70 | **73.43** | 95.01 | 82.38 | 80.79 | 70.88 | 92.29 | 87.94 | 88.01 | 84.49 | 8.28 | 0.7930 |
| TMSA-Net (2025)[**] | 87.50 | 64.24 | 96.18 | 84.03 | 79.86 | 67.71 | 93.06 | 90.79 | 85.42 | 83.20 | 10.94 | 0.7762 |
| **EEG-CSANet** | **94.44** | 71.88 | **98.26** | **91.32** | **84.03** | **79.51** | 93.75 | **92.36** | **91.32** | **88.54** | 8.41 | **0.8472** |

Table 3: Performance comparison on BCIC-IV-2b.

| Methods (Year) | B1 | B2 | B3 | B4 | B5 | B6 | B7 | B8 | B9 | Acc | Std | Kappa |
|---|---|---|---|---|---|---|---|---|---|---|---|---|
| EEGNet (2018)[**] | 75.94 | 57.64 | 58.43 | 98.13 | 81.25 | 88.75 | 84.06 | 93.44 | 89.69 | 80.81 | 14.46 | 0.6096 |
| EEG-Conformer (2022)[*] | 82.50 | 65.71 | 63.75 | 98.44 | 86.56 | 90.31 | 87.81 | 94.38 | 92.19 | 84.63 | 12.18 | 0.6926 |
| ATCNet (2022)[*] | 79.38 | 75.00 | **88.75** | 98.13 | 96.56 | 88.13 | 94.38 | 94.69 | 85.31 | 88.93 | **7.94** | 0.7785 |
| ADFCNN (2024)[**] | 78.13 | 72.14 | 87.50 | 98.44 | 97.81 | 90.00 | 92.81 | 92.81 | 90.63 | 88.92 | 8.69 | 0.7749 |
| EISATC-Fusion (2024)[**] | 75.00 | 72.86 | 86.56 | 96.88 | 97.81 | 84.38 | 94.06 | 93.75 | 87.58 | 87.58 | 9.07 | 0.7515 |
| EEG-TransNet (2024)[**] | 79.06 | 70.71 | 87.81 | 98.44 | 96.88 | **91.56** | 91.88 | 95.63 | 90.00 | 89.11 | 8.98 | 0.7822 |
| MSTFNet (2024)[*] | 82.72 | 73.26 | 82.34 | **98.75** | 97.44 | 90.86 | 92.51 | 94.52 | 90.99 | 89.27 | 8.27 | 0.7800 |
| TMSA-Net (2025)[*] | 82.81 | 71.07 | 87.81 | 98.13 | 98.44 | **91.56** | 94.38 | 95.63 | 87.50 | 89.70 | 8.74 | 0.7940 |
| **EEG-CSANet** | **83.13** | **73.57** | 88.13 | 99.38 | **99.69** | 91.25 | **95.31** | **96.88** | **92.50** | **91.09** | 8.48 | **0.8218** |

**Subject-dependent Experiment.** We present all experimental results in Tables 2, 3 and 4, where bold values indicate the best performance among all methods. Our model achieved higher classi-

Table 4: Performance comparison on HGD and BCIC-IV-2a.

| Methods (Year) | HGD (Subject-dependent) | | BCIC-IV-2a (Subject-independent) | |
|---|---|---|---|---|
| | Acc | Kappa | Acc | Kappa |
| EEGNet (2018) | 88.87 | 0.8531 | 58.19 | 0.4425 |
| EEG-Conformer (2022) | 91.92 | 0.8824 | – | – |
| ATCNet (2022) | 95.54 | 0.9405 | 62.85 | 0.5047 |
| EEG-TransNet (2024) | 94.82 | 0.9309 | 65.73 | 0.5430 |
| ADFCNN (2024) | 95.17 | 0.9381 | – | – |
| MCMTNet (2025) | 95.73 | 0.9430 | 66.31 | 0.5508 |
| TMSA-Net (2025) | 95.90 | 0.9452 | – | – |
| **EEG-CSANet** | **97.15** | **0.9627** | **69.68** | **0.5957** |

fication accuracy and Kappa coefficients on the BCIC-IV-2a, BCIC-IV-2b, and HGD datasets, outperforming existing SOTA methods. On BCIC-IV-2a and BCIC-IV-2b, paired-sample t-tests further confirmed the significant differences in decoding performance between our method and other models. On the HGD dataset, our method achieved near-saturated performance across multiple subjects while maintaining low variance.

Existing SOTA models (Song et al., 2022) (Altaheri et al., 2022) (Tao et al., 2023) (Ma et al., 2024) (Jin et al., 2024) (Liang et al., 2024) (Yang et al., 2025) (Zhao & Zhu, 2025) usually focus on the integration of attention mechanisms and convolution. In contrast, EEG-CSANet also leverages the integration of multi-branch feature extraction, multiscale pooling, and dual Top-k attention, which enhances discriminative information selection and effectively captures both local and global dependencies. In addition, the residual TCN block strengthens the resilience of temporal feature representations. Through this complementary design, the proposed model achieves robust and generalizable decoding performance across datasets and subjects.

**Subject-independent Experiment.** The comparative analysis of subject-independent classification performance between EEG-CSANet and several SOTA methods on BCIC-IV-2a is detailed in Table 4. Evidently, EEG-CSANet achieved superior classification performance, thereby effectively validating its transferability.

## 3.4 ABLATION STUDY

To investigate the effects of different components in the proposed model, we conducted ablation experiments on BCIC-IV-2a for the S&R, TCN, Residual, and MSCA modules. In addition, for the MSCA module, we separately evaluated the contributions of its two main components: Top-k and Multiscale-AvgPool. We analyzed the results using three metrics: mean accuracy, standard deviation, and Kappa coefficient.

As shown in Table 6, among the four modules, the S&R module provides the most significant improvement in classification performance. This phenomenon may be attributed to the relatively small amount of training data in the BCIC-IV-2a, which makes complex models more prone to overfitting during training, whereas S&R can effectively mitigate this issue to some extent. Among the remaining modules, the performance improvement second only comes from the residual connection. This indicates that the MSCA operation may lead to the loss of some temporal feature information, and the residual connection can partially preserve and supplement this information, thereby enhancing the representation of global features.

The experimental results further show that removing any module from the model leads to a varying degree of performance degradation. This demonstrates that all modules play indispensable roles in the overall architecture, validating the necessity of their design.

## 4 DATA VISUALIZATION

### 4.1 CONFUSION MATRIX

The confusion matrix is a fundamental tool for evaluating classification models, providing insights into accuracy, error patterns, and class-specific biases for a comprehensive performance assess-

ment. We employed confusion matrices to examine the performance of EEG-CSANet across three datasets. To provide a thorough assessment of the model's overall classification performance, we aggregated the predictions of all subjects on the test sets and constructed confusion matrices based on this combined data. Figure 2 shows that EEG-CSANet achieved relatively balanced classification performance across all tasks. Specifically, on BCIC-IV-2A, the classification of tongue movement (task 3) was slightly worse than other classes, while on HGD, the classification performance for left-hand (task 0) and right-hand (task 1) movements was slightly lower compared to the other tasks.

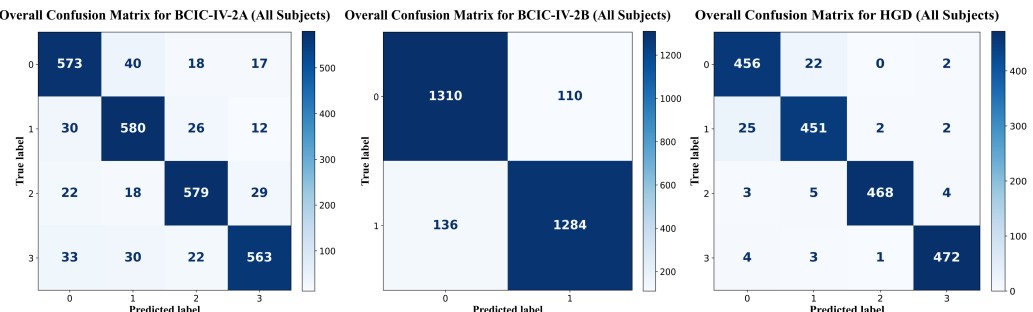

Figure 2: Confusion matrix of EEG-CSANet on the three datasets.

## 4.2 UMAP

Uniform Manifold Approximation and Projection (UMAP) is a nonlinear dimensionality reduction and visualization method that preserves local structure while capturing global topology. Compared with t-SNE, UMAP offers higher computational efficiency and scalability for high-dimensional data, revealing clustering patterns and distribution characteristics in low-dimensional space (McInnes et al., 2018). We applied UMAP to visualize features extracted by the four branches of EEG-CSANet. Figure 3 indicate distinct feature representations across branches, reflecting diverse characterizations of the input signals. By integrating these multiscale features, the model can more comprehensively capture and leverage information at different granularities.

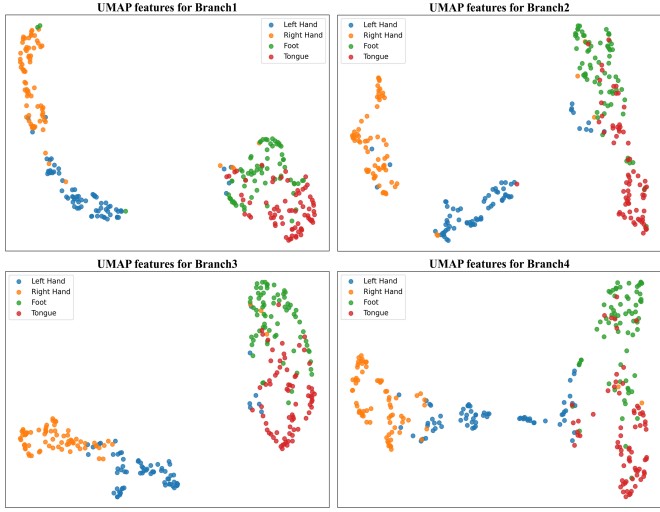

Figure 3: UMAP of EEG-CSANet on four brenches.

## 4.3 VISUALIZATION OF CONVOLUTIONAL FEATURES

To further investigate the features extracted by the four branches using convolutional kernels of different sizes, we applied the welch method to compute the power spectral density (PSD) of the EEG

signals after temporal convolution. From top to bottom, convolutional kernel sizes of 64, 32, 16, and 8 were used. In Figure 6, the blue line represents the original signal, while the red line represents the convolved signal. Figure 6.a shows a pronounced energy enhancement in the Theta, Alpha, and Beta bands, particularly in the 8–13 Hz and 13–20 Hz ranges, whereas the Delta band exhibits a marked decrease in energy. Figure 6.b exhibits a sharp peak in the Alpha band, with a notable increase in the Delta band, but minimal enhancement in the high-frequency range. Figure 6.c is similar to Figure 6.b but shows better representation in the transition from Alpha to Beta. Figure 6.d displays relatively weaker low-frequency activity (0.5–13 Hz) compared to Figure 6.b and Figure 6.c, but shows marked enhancement in the high-frequency Beta to Gamma (13–50 Hz) range, with a peak around 20 Hz.

This may be attributed to the larger receptive field of the large kernels, which can capture the overall slow-varying trends of the signal and cross-frequency rhythmic information due to their coverage of longer temporal windows. Conversely, smaller kernels focus on local, rapidly varying signal patterns, making them more sensitive to high-frequency features of EEG signals, which often reflect transient, short-duration neural activity.

Therefore, the multi-branch temporal convolutional module indeed captures features from different frequency bands, and features in different frequency bands often correspond to distinct spatial distribution patterns. **This further validates the advantage of our multi-branch architecture**: by separately extracting multi-scale features along the temporal dimension while preserving their independent spatial representation pathways, it effectively mitigates the issue of spatial information confusion or loss that arises in traditional approaches when multi-scale features are directly concatenated.

### 4.4 Comparison Between Main–Auxiliary and Hierarchical Structures

In our original model, the Query of each MSCA branch was derived from the features obtained after the DW-Spa-Conv module in the first branch. Inspired by (Liu et al., 2025a), we compared the performance of the original main–auxiliary structure with a hierarchical structure,as is shown in Figure 5 and Appendix A.6. In the hierarchical design, the features extracted after the DW-Spa-Conv module in the first branch serve as the Query for the second branch's MSCA, whose output is then passed sequentially as the Query to the third and fourth branches. This layer-by-layer propagation enables each branch to attend not only to its own representations but also to inherit information from the preceding branch, thereby achieving cross-branch multi-level feature fusion. As is shown in Table 7, compared with main–auxiliary structure, the Hierarchical structure yields slightly lower classification performance, though the difference is not statistically significant according to t-tests. To present the best-performing model, we ultimately adopted the main–auxiliary structure.

## 5 Conclusion

In this paper, we propose EEG-CSANet, a multi-branch feature fusion framework for MI classification. To address the spatial information loss caused by shared weights in multiscale feature extraction, EEG-CSANet introduces a parallel design that assigns an independent spatial module to each temporal scale, enabling more precise characterization of the spatiotemporal heterogeneity of EEG signals. For multi-branch feature fusion, EEG-CSANet adopts a main–auxiliary collaborative architecture: the main branch leverages multiscale self-attention to model core spatiotemporal patterns, while the auxiliary branch employs multiscale sparse cross-attention to enable efficient local interactions with the main branch. Experimental results demonstrate that EEG-CSANet achieves SOTA performance across three public MI datasets. Moving forward, we expect EEG decoding to to be more closely aligned with neural mechanism-driven strategies, fostering the design of methods specifically adapted to the properties of EEG signals, which in turn can improve the physiological plausibility and generalizability of EEG-based models.

## 6 Acknowledgments

We thank the large language models for their assistance in language editing and expression refinement of this manuscript.

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

# A APPENDIX

## A.1 DATASETS DISCRIPTION AND PREPROCESSION

**BCIC-IV-2a** (Brunner et al., 2008) provided by the Graz University of Technology, consists of recordings from nine healthy subjects. Each subject performed four MI tasks, including left hand, right hand, foot, and tongue movements. EEG signals were recorded using 22 Ag/AgCl electrodes at a sampling rate of 250 Hz. For each subject, two sessions were recorded on different days, serving as the training and testing sets, respectively. Each session comprises 288 trials, i.e., 72 trials per task. In our study, a 4-second time window was adopted for each trial (Song et al., 2020).

**BCIC-IV-2b** (Leeb et al., 2008) also includes recordings from nine healthy subjects. Each subject performed two MI tasks, namely left-hand and right-hand movements. EEG signals were recorded using three bipolar electrodes (C3, Cz, and C4) at a sampling rate of 250 Hz. A 4-second time window was adopted for each trial (Song et al., 2020). Each subject completed five sessions: the first two sessions without feedback, each containing 120 trials, and the subsequent three sessions with feedback, each containing 160 trials. In our experiments, the first three sessions were used for training and the last two sessions for testing, as done in (Ma et al., 2024).

**HGD** (Schirrmeister et al., 2017) consists of recordings from 14 healthy subjects. Each subject performed four MI tasks, including left-hand, right-hand, both-feet movements, and rest (where a visual cue identical to that of the other tasks was presented on the screen, but subjects were instructed not to perform any movement). The original EEG signals were recorded using 128 channels at a sampling rate of 500 Hz, with a 4-second time window for each trial. Following (Zhao & Zhu, 2025), we selected 44 channels covering the motor cortex for EEG decoding, downsampled the signals to 250 Hz, and applied standard normalization. For each subject, the dataset includes both training and testing sessions: the training set comprises approximately 880 trials per subject, while the testing set contains about 160 trials per subject.

## A.2 BASELINES

**EEGNet** (Lawhern et al., 2018) is a lightweight convolutional neural network specifically tailored for EEG signal classification. By leveraging depthwise separable convolutions, it achieves an effective trade-off between performance and computational efficiency. Its architecture is simple, compact, and easy to implement or transfer across different tasks. Owing to these advantages, EEGNet has been widely adopted in various BCI applications, such as MI and Event-Related Potention decoding, and has become one of the benchmark models in deep learning for EEG analysis.

**EEG-Conformer** (Song et al., 2022) is a deep learning model specifically developed for EEG signal classification, which integrates CNNs with a Transformer architecture. In this framework, CNNs are employed to extract local spatiotemporal features, while the Transformer is utilized to model long-range temporal dependencies and global contextual information. By effectively combining the strengths of feature extraction and sequence modeling, EEG-Conformer provides a powerful and comprehensive representation of EEG data.

**ATCNet** (Altaheri et al., 2022) is a compact and interpretable attention-based temporal convolutional network specifically designed for EEG motor imagery classification. It integrates principles of scientific machine learning by employing convolutional layers to extract spatiotemporal features, while multi-head self-attention mechanisms emphasize the most informative temporal segments. In addition, a TCN is incorporated to capture long-term temporal dependencies. With its concise architecture and strong interpretability, ATCNet demonstrates improved decoding performance for EEG signals.

**ADFCNN** (Tao et al., 2023) is a deep learning model tailored for MI–BCI. It adopts a dual-scale convolutional architecture to separately extract temporal features of EEG rhythms as well as global and fine-grained spatial features. To further enhance representation learning, a self-attention mechanism is introduced, enabling the model to dynamically weight and fuse multiscale information according to intrinsic feature similarities, thereby improving feature discriminability. This design effectively overcomes the limitations of traditional single-scale or traditional multiscale CNNs in feature extraction and fusion, offering stronger modeling capabilities for both spectral and spatial information.

**EEG-Transnet** (Ma et al., 2024) is a MI-EEG classification model that integrates CNNs with a self-attention mechanism. The model extracts multimodal temporal features from both the mean and variance dimensions, while a shared self-attention module captures global temporal dependencies. A convolutional encoder is then employed to fuse these features, thereby enhancing their discriminability. In addition, a signal S&R-based data augmentation strategy is introduced to further improve robustness and decoding performance.

**MSTFNet** (Jin et al., 2024) is an end-to-end convolutional neural network proposed for motor imagery EEG classification. To overcome the limitations of traditional single-scale convolutions in feature extraction, MSTFNet is designed with four core modules: feature enhancement, multiscale temporal feature extraction, spatial feature extraction, and feature fusion. Through the collaborative extraction of multiscale spatiotemporal features and the adoption of refined fusion strategies, the model substantially improves the decoding of EEG signals.

**EISATC-Fusion** (Liang et al., 2024) is a neural network model tailored to MI-EEG decoding, which integrates Inception modules, multi-head self-attention, TCNs, and a layer fusion structure. Specifically, the model employs DS-Inception to extract multiscale frequency band features, incorporates cnnCosMSA to alleviate attention collapse and enhance interpretability, and applies depthwise separable convolutions to reduce parameter complexity. Furthermore, by combining both feature-level and decision-level fusion strategies, EISATC-Fusion achieves improved robustness in EEG signal decoding.

**MCMTNet** (Yang et al., 2025) is a deep learning model capable of directly processing raw EEG signals without the need for complex preprocessing. Its architecture is composed of three main components: a multi-domain convolution module, a multi-head attention module, and a TCN. The multi-domain convolution module is responsible for filtering and feature extraction, the multi-head attention module enhances cross-scale correlations, and the TCN improves temporal coherence.

**TMSA-Net** (Zhao & Zhu, 2025) is a neural network model that integrates CNNs with an improved Transformer-based attention mechanism. The model first employs CNNs to extract local spatiotemporal features, and then introduces a novel attention module to enhance global modeling capability across both the channel and temporal dimensions, thereby effectively bridging the gap between local and global representations. By optimizing the attention structure, TMSA-Net not only reduces computational overhead but also improves feature fusion efficiency, enhancing the model's sensitivity to key EEG patterns and its interpretability.

## A.3 EVALUATE METRICS

**Average Accuracy** provides a holistic measure of model performance by averaging classification accuracy over all categories. Given an $n$-class problem, the average accuracy (Acc) can be expressed as:

$$\text{Acc} = \frac{\sum_{i=1}^{n} \text{TP}_i}{\sum_{i=1}^{n} (\text{TP}_i + \text{FP}_i)} \tag{12}$$

where $\text{TP}_i$ and $\text{FP}_i$ denote the number of true positives and false positives for class $i$, respectively.

**Kappa coefficient** is employed to evaluate the agreement between predicted and true labels, accounting for the agreement occurring by chance. It is especially effective evaluating model performance under class-imbalanced conditions.

$$\kappa = \frac{p_o - p_e}{1 - p_e} \tag{13}$$

where $p_o$ denotes the observed agreement and $p_e$ is the expected agreement under random guessing.

## A.4 EXPERIMENT RESULTS

Table 5 reports subject-dependent performance comparisons on HGD, we present the comparison results of all subjects with SOTA models.

Table 6 presents the ablation experiments on BCIC-IV-2a for the S&R, TCN, Residual, and MSCA modules. As to MSCA, we separately evaluated the contributions of its two main components: Top-k and AvgPool.

Table 5: Performance comparison on HGD.

| Subjects | EEGNet | EEG-Conformer | ATCNet | EEG-TransNet | ADFCNN | MCMTNet | TMSA-Net | EEG-CSANet |
|---|---|---|---|---|---|---|---|---|
| H01 | 85.15 | 92.28 | 95.00 | 94.38 | 90.00 | 95.69 | **98.13** | 97.50 |
| H02 | 85.02 | 89.78 | **97.50** | 95.00 | 92.50 | 97.00 | 94.38 | **97.50** |
| H03 | 98.82 | 98.23 | 99.38 | 99.38 | 99.17 | 99.75 | 99.38 | **100.00** |
| H04 | 95.70 | 98.85 | 98.75 | 99.38 | **100.00** | 98.31 | **100.00** | 99.38 |
| H05 | 93.20 | 90.73 | 97.50 | 94.38 | **100.00** | 97.94 | **100.00** | 99.38 |
| H06 | 90.12 | 93.33 | 95.00 | 95.00 | 98.13 | 97.81 | **98.75** | **98.75** |
| H07 | 86.28 | 90.04 | 94.38 | 94.97 | 92.50 | 93.77 | **98.75** | 95.63 |
| H08 | 88.82 | 85.10 | 96.88 | 94.38 | **99.38** | 95.87 | 96.88 | 96.88 |
| H09 | 95.07 | 98.23 | 98.13 | 97.50 | **100.00** | 97.62 | 99.38 | 99.38 |
| H10 | 88.22 | 90.73 | 91.88 | 95.63 | 96.25 | 93.81 | 95.00 | 96.25 |
| H11 | 76.35 | 79.58 | 85.63 | 83.13 | 98.13 | 88.88 | 98.75 | 92.63 |
| H12 | 96.95 | 96.45 | 98.13 | 98.75 | 98.13 | 98.06 | 98.75 | **98.75** |
| H13 | 83.75 | 92.65 | 95.00 | 95.60 | 97.50 | 96.10 | 97.50 | **98.75** |
| H14 | 80.72 | 82.05 | **94.38** | 90.00 | 70.63 | 89.56 | 66.88 | 89.38 |
| Acc | 88.87 | 91.29 | 95.54 | 94.82 | 95.17 | 95.73 | 95.90 | **97.15** |
| Std | 6.53 | 5.90 | 3.54 | 4.17 | 7.74 | 3.23 | 8.52 | **2.97** |
| Kappa | 0.8531 | 0.8824 | 0.9405 | 0.9309 | 0.9381 | 0.9430 | 0.9452 | **0.9627** |

Table 6: Ablation study of different modules.

| S&R | TCN | Residual | MSCA | | Acc | Std | Kappa |
|---|---|---|---|---|---|---|---|
| | | | Top-k | AvgPool | | | |
| ✓ | ✓ | ✓ | ✓ | ✓ | 88.54 | 8.41 | 0.8472 |
| × | ✓ | ✓ | ✓ | ✓ | 81.35 | 7.91 | 0.7514 |
| ✓ | × | ✓ | ✓ | ✓ | 87.61 | 8.09 | 0.8348 |
| ✓ | ✓ | × | ✓ | ✓ | 86.38 | 9.21 | 0.8184 |
| ✓ | ✓ | ✓ | × | × | 86.78 | 8.98 | 0.8224 |
| ✓ | ✓ | ✓ | ✓ | × | 87.50 | 9.74 | 0.8334 |
| ✓ | ✓ | ✓ | × | ✓ | 87.73 | 8.23 | 0.8364 |

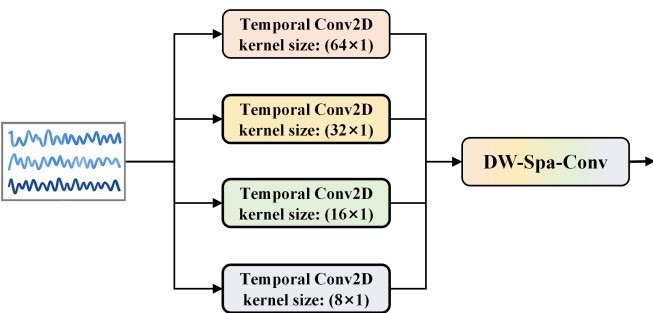

Figure 4: Traditional multiscale temporal feature extraction methods..

## A.5 VISUALIZATION OF CONVOLUTIONAL FEATURE

To examine the features extracted by the four branches with different convolutional kernel sizes (64, 32, 16, and 8), we employed the welch method to compute the EEG PSD. In Figure 6, the blue and red lines denote the original and convolved signals, respectively.

## A.6 HIERARCHICAL GUIDANCE OF STRUCTURE

As shown in Figure 5, in the hierarchical design, the features extracted by the DW-Spa-Conv module in the first branch are used as the Query for the MSCA in the second branch. Its output is then sequentially passed as the Query to the third and fourth branches. Through this layer-by-layer propagation mechanism, each branch can not only focus on its own representations but also inherit information from the preceding branch, thereby enabling cross-branch multi-level feature fusion. Table 7 presents a detailed comparison of the classification accuracy between Main–auxiliary Guidance and Hierarchical Guidance.

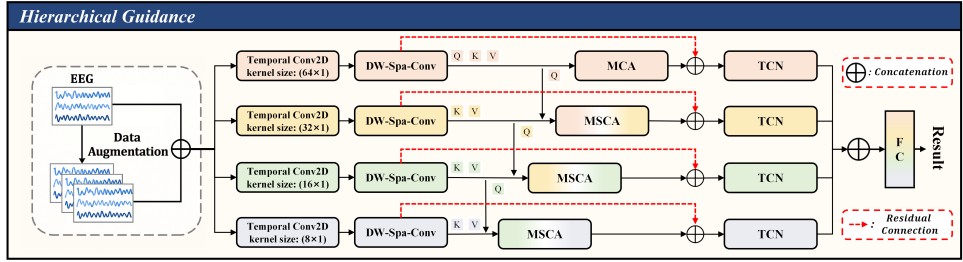

Figure 5: Hierarchical Guidance of Structure.

Table 7: Comparison between Main–auxiliary Guidance and Hierarchical Guidance.

| Dataset | Main–auxiliary Guidance | | | Hierarchical Guidance | | |
|---|---|---|---|---|---|---|
| | ACC | Std | Kappa | ACC | Std | Kappa |
| BCIC-IV-2a | 88.54 | 8.31 | 0.8472 | 88.08 | 8.77 | 0.8411 |
| BCIC-IV-2b | 91.09 | 8.48 | 0.8218 | 90.89 | 8.57 | 0.8178 |
| HGD | 96.43 | 4.52 | 0.9542 | 96.29 | 4.01 | 0.9503 |

## A.7 HYPERPARAMETERS

Table 8 provides a detailed summary of the hyperparameters for each module of the model.

Table 8: Hyperparameters of different modules.

| Module | Hyperparameters | Settings |
|---|---|---|
| Data Augmentation | Segment ($S$) | 8 |
| Temporal Conv2D | Kernel Size ($K_1, K_2, K_3, K_4$) | (64, 32, 16, 8) |
| | Filters ($F_1, F_2, F_3, F_4$) | (16, 16, 16, 16) |
| DW-Spa-Conv | DW Kernel Size ($K_5$) | (C, 1) |
| | DW Filters ($F_5$) | 16 |
| | Depth multiplier ($D$) | 2 |
| | Pooling Size ($P_1, P_2$) | (8, 7) |
| | Spa Filters ($F_6$) | 32 |
| | Spa Filters ($K_6$) | 32 |
| | Dropout | 0.5 |
| MSCA | Pooling Size ($Q_1, Q_2, Q_3$) | (3, 5, 7) |
| | Pooling Padding ($G_1, G_2, G_3$) | (1, 2, 3) |
| | Number of Heads ($h$) | 8 |
| | Top-k ($k_1, k_2$) | (2, 3) |
| TCN | Dilation factor ($d_1, d_2$) | (1, 2) |
| | Kernel size ($K_t$) | 4 |
| | Filters ($F_t$) | 32 |
| | Dropout | 0.3 |

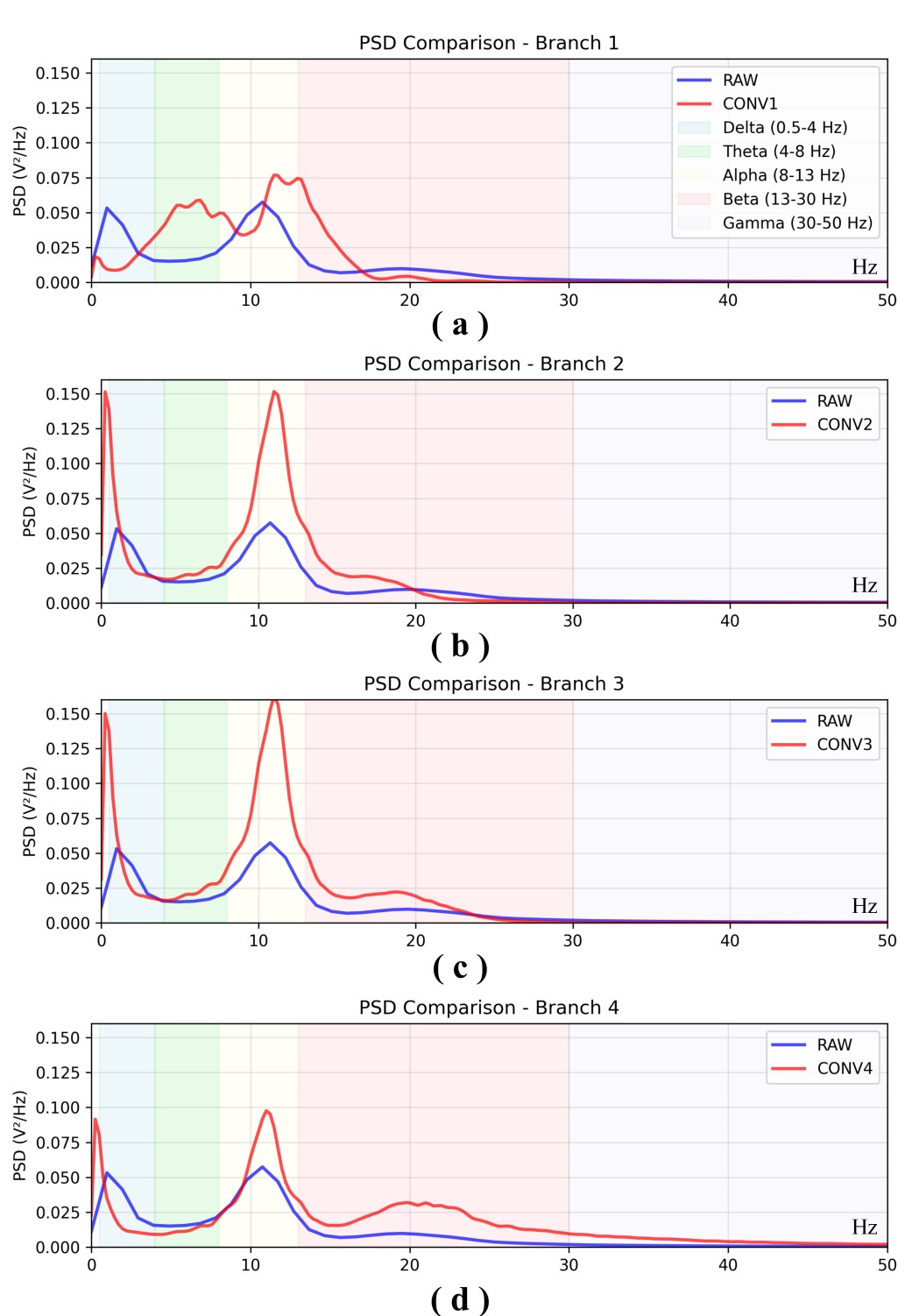

Figure 6: PSD of EEG-CSANet on four branches (HGD: Sub6).

