# OpenReview forum: "Fusion of Multiscale Features via Centralized Sparse-attention Network for EEG Motor Imagery Classification"
_ICLR.cc/2026/Conference — ICLR 2026 Conference Withdrawn Submission_

### Official Review · Reviewer_4vAd · 2025-10-25

**Soundness:** 2
**Presentation:** 3
**Contribution:** 1
**Rating:** 0
**Confidence:** 5

**Summary:**

The authors designed a new model network for multi-branch multiscale feature fusion. Specifically, global spatiotemporal patterns as well as local interactions are modelled for EEG signals. However, the technical innovation of this work is limited.

**Strengths:**

[1] Multi-branch feature extraction framework, but not new in the EEG MI field\
[2] Multiscale feature extraction, also not new in the EEG MI field

**Weaknesses:**

[1] The technical contributions are limited, except for the network design.\
[2] Data pre-processing – The authors should describe how they pre-process the EEG signals.\
[3] Datasets – The authors are encouraged to use the largest EEG MI dataset for more solid experiments and model evaluation:\
EEG Motor Movement/Imagery Dataset\
Link: https://archive.physionet.org/pn4/eegmmidb/

**Questions:**

[1] What exactly are the technical innovations of the manuscript? Or, what is unique and new in the field of EEG MI?\
[2] Did the authors have any explainability method to explain the model’s behavior?\
[3] What are the contributions of different scales/branches to the final model performance?\
[4] Are there any ablation studies on the Top-k sparsification operation?

---

### Official Review · Reviewer_EZwK · 2025-10-31

**Soundness:** 2
**Presentation:** 3
**Contribution:** 1
**Rating:** 2
**Confidence:** 4

**Summary:**

This paper proposes a multi-branch parallel framework for motor imagery EEG classification called EEG-CSANet. Each time-scale branch independently extracts spatial features. The main branch employs multi-head self-attention, while the auxiliary branches use multi-scale sparse cross-attention for feature fusion. The authors validated the framework on three public datasets (BCI-IV-2a/2b and HGD) and reported performance superior to recent methods.

**Strengths:**

1. **Clear architectural motivation:** The paper points out that previous approaches following the “multi-scale temporal convolution -> unified spatial module” paradigm implicitly assume spatial homogeneity, which may overlook spatial heterogeneity across different frequency bands or temporal scales.

2. **The experimental process is well-standardized:** The proposed method outperforms recent approaches on all three datasets—BCI-IV-2a, BCI-IV-2b, and HGD—and also reports results under subject-independent scenarios. The visualizations including confusion matrices, UMAP, and PSD maps, strengthens the persuasiveness and interpretability of the results.

3. **The paper is well-written and logically coherent.**

**Weaknesses:**

1. **Lack of substantial innovation.** The proposed method offers limited novelty. The so-called “main–auxiliary collaborative fusion” essentially represents a minor modification of the MCMTNet framework proposed by Yang et al. (2025). Overall, the design still follows the conventional CNN + Attention + TCN hybrid framework, such as:

&emsp;&emsp;[1] MCMTNet: Advanced network architectures for EEG-based motor imagery classification

&emsp;&emsp;[2] TMSA-Net:A novel attention mechanism for improved motor imagery EEG signal processing

&emsp;&emsp;[3] Attention-based convolutional neural network with multi-modal temporal information fusion for motor imagery EEG decoding

&emsp;&emsp;[4] EISATC-Fusion: Inception Self-Attention Temporal Convolutional Network Fusion for Motor Imagery EEG Decoding

2. **Outdated technical approach.** The combination of multi-scale convolution and attention-based fusion has become a standard practice in recent years. The paper does not introduce any new modeling mechanisms—such as graph signal modeling, time–frequency alignment, or enhanced interpretability frameworks. Instead, it merely makes minor adjustments to the fusion strategy, resulting in limited methodological contribution.

3. **Insufficient statistical validation.** Although the paper claims to outperform state-of-the-art (SOTA) methods, the reported improvements in Tables 2-4 are mostly within the 2-3% range. However, the authors do not provide full statistical evidence, such as p-values or analysis of variance (ANOVA) results, making it impossible to confirm the statistical significance of the reported gains.

4. **Insufficient experimental validation.** The experiments lack a comprehensive evaluation of parameter sensitivity—for example, variations in the number of branches or Top-k ratios. Moreover, the study omits any systematic analysis of model efficiency, such as FLOPs or inference latency. Additionally, the comparisons with models without data augmentation are not fully fair, further limiting the credibility and generalization of the experimental results.

5. **Writing errors.** There are several noticeable editorial issues in the paper. Figure 1, Table 3, and Table 5 contain labeling errors.

**Questions:**

1. Given the high structural overlap with MCMTNet, what is the primary source of the reported performance improvement?

2. Would the performance change significantly if the Top-k sparsification module were removed?

3. The paper mentions that the data augmentation operations must be executed independently within each training batch and then merged with the original data before being fed into the model. How does this approach compare in terms of computational efficiency with models that do not require complex preprocessing?

---

### Official Review · Reviewer_agDh · 2025-11-01

**Soundness:** 2
**Presentation:** 2
**Contribution:** 1
**Rating:** 2
**Confidence:** 4

**Summary:**

This paper proposes EEG-CSANet, a multi-scale, multi-branch network for EEG-based motor imagery classification that captures both temporal and spatial diversity of brain signals. Each branch extracts temporal features at a distinct scale and applies its own spatial convolution, allowing the model to learn scale(frequency)-specific spatial patterns and better represent the spatiotemporal characteristics of EEG data. The EEG-CSANet also utilizes auxiliary branch that applies multiscale sparse cross-attention, facilitating efficient local feature interactions with the main (global) branch. EEG-CSANet is evaluated on three public MI-EEG datasets, achieving state-of-the-art performance.

**Strengths:**

The paper effectively addresses a key limitation of previous multiscale EEG models by separating spatial representation learning across different temporal scales (frequency bands), allowing the network to better capture EEG’s spatial, spectral, and temporal characteristics. The framework is coherently organized and experimentally verified, with results showing that each branch focuses on distinct EEG feature patterns.

**Weaknesses:**

The paper’s contribution is limited, as the proposed model mainly combines existing concepts such as multiscale temporal branches, spatial convolutions, and attention-based fusion. While the design is coherent, it lacks methodological novelty beyond integrating these well-established components.

The paper provides little justification for various hyperparameter selections. For instance, why were only four temporal kernel sizes (64, 32, 16, 8) used? Similarly, in the Top-k sparsification operation, why were only two k values chosen? The authors should explain the rationale behind these specific design choices, as well as whether they were empirically optimized or heuristically determined.

The experimental design appears insufficiently rigorous. Both the subject-independent evaluation and the ablation studies were conducted on a single dataset, and statistical tests (e.g., t-test) were not consistently applied across experiments.

**Questions:**

In many existing MI-EEG encoders (e.g., EEGNet), temporal convolution is followed by depthwise convolution to apply distinct spatial filters to each temporal kernel. This extracts spectral-temporal features from different frequency bands via randomly initialized temporal kernels, and then learns frequency-specific spatial patterns through depthwise spatial convolution. The proposed approach (independent spatial feature extraction) appears conceptually similar to this widely adopted design. The authors should clarify in detail how their method differs from such conventional temporal-then-spatial convolutional pipelines, both in terms of architecture and functional motivation.

Numerous previous studies have also adopted multi-branch architectures [1],[2]. In particular, [1] employs multiple branches to learn spatial and spectral-temporal representations from different frequency bands and an additional global branch to capture comprehensive (global) MI-related patterns. The authors should explicitly discuss how their method is differentiated from these prior multi-branch models.
[1] Kim et al. "A learnable continuous wavelet-based multi-branch attentive convolutional neural network for spatio–spectral–temporal EEG signal decoding." Expert Systems with Applications 251 (2024): 123975.
[2] Zhao et al. "A multi-branch 3D convolutional neural network for EEG-based motor imagery classification." IEEE transactions on neural systems and rehabilitation engineering 27.10 (2019): 2164-2177.

The paper only reports subject-independent (cross-subject) evaluation results for the BCIC-IV-2a dataset. Could the authors clarify why this setting was not applied to the other datasets (BCIC-IV-2b and HGD)?

In Table 6, removing S&R augmentation method causes a large drop in accuracy, while Table 2 shows that the model without augmentation performs worse than several baselines. This raises the question of whether the performance gain primarily comes from the proposed architecture or from the augmentation. Did the compared baselines also use the same augmentation strategy? Moreover, were ablation studies conducted on the other datasets to confirm that the improvement is not dataset-specific?

The ablation results show only minor numerical differences. Statistical significance testing (e.g., paired t-test, Wilcoxon signed-rank test) would be necessary to support the claim that each component contributes meaningfully to performance improvement.

The authors claim that multiple filters capture different frequency bands, yet only four filters are used. Please justify why four were chosen and whether increasing the number of filters could yield more fine-grained spectral representations.

The PSD visualizations for kernel sizes 32 and 16 appear quite similar, suggesting limited diversity in learned spectral information. Why did the authors select kernel sizes of 64, 32, 16, and 8? Additionally, how do these differ from randomly initialized kernels?

Fig. 1 could be improved by explicitly labeling the main and auxiliary branches to clarify data flow and attention interactions, which would make the architecture easier to follow.

---

### Official Review · Reviewer_Xjjz · 2025-11-01

**Soundness:** 2
**Presentation:** 2
**Contribution:** 2
**Rating:** 2
**Confidence:** 5

**Summary:**

This paper presents EEG-CSANet, a novel multi-branch feature fusion framework designed for Motor Imagery EEG classification. The core innovation addresses the limitation in conventional multiscale temporal feature extraction, where combining features before spatial modeling overlooks the inherent spatiotemporal heterogeneity of EEG signals.

**Strengths:**

1.	The Main-Auxiliary Collaborative Fusion with MSCA is an efficient mechanism. The combination of multiscale pooling and Top-k sparsification in the cross-attention effectively balances the need for global context with the preservation of local details, while explicitly addressing noise.
2.	The model achieves superior results compared to recent SOTA methods across three MI-EEG datasets, including strong performance in the challenging subject-independent experiment. This demonstrates both the effectiveness and transferability of the model.

**Weaknesses:**

1.	Poor structural organization of the paper: the authors frequently reference figures and tables located in the Appendix while describing and interpreting them in the main text. This creates a confusing reading experience. For example, Figure 6 is referenced and discussed on page 9 but appears on page 17; Figure 5 and Table 7 are discussed on page 9 but appear on page 16. If these figures contain key experimental results, they should be placed within the main body rather than in the Appendix. A well-structured paper should be self-contained, with critical results presented directly in the main text.

2.	While the overall model design is coherent and complete, the novelty is limited. The approach does not introduce a fundamentally new method for spatiotemporal feature extraction in EEG, and the multiscale fusion strategy remains fairly conventional.

3.	The core claim that each temporal scale should have its own spatial processing pathway is not sufficiently validated experimentally. Although the ablation studies test the contribution of several modules, they do not analyze cross-branch interactions (e.g., varying the number or configuration of auxiliary branches). These analyses are essential to substantiate the main hypothesis.

4.	The ablation results show that the Signal Segmentation and Reconstruction (S&R) data augmentation contributes the most to classification improvement—removing S&R reduces accuracy from 88.54% to 81.35%. In contrast, removing other modules yields only minor changes. This raises questions: does this indicate that preprocessing is more critical than the proposed architecture itself? Why do other methods that also employ S&R or similar augmentations not achieve such large gains?

5.	The study focuses only on MI datasets. Evaluating EEG-CSANet on other paradigms (e.g., emotion recognition, ERP tasks) would strengthen the generality and robustness claims.

6.	While the PSD visualizations help illustrate feature behavior, more quantitative neurophysiological validation (e.g., frequency-band activation analysis) would add depth. Furthermore, it remains unclear whether the PSD analysis actually verifies that the model captures multiscale spatial characteristics—how do the authors define “spatial features” in this context?

**Questions:**

Please refer to the Weaknesses section for detailed questions and suggestions to the authors.

---

### Note · Authors · 2025-11-18

I have read and agree with the venue's withdrawal policy on behalf of myself and my co-authors.